# Sensibility and Specificity of the VitaPCR™ SARS-CoV-2 Assay for the Rapid Diagnosis of COVID-19 in Older Adults in the Emergency Department

**DOI:** 10.3390/v15010189

**Published:** 2023-01-09

**Authors:** Francesco Piacenza, Antonio Cherubini, Roberta Galeazzi, Maurizio Cardelli, Robertina Giacconi, Elisa Pierpaoli, Francesca Marchegiani, Fiorella Marcheselli, Rina Recchioni, Tiziana Casoli, Elisabetta Farnocchia, Beatrice Bartozzi, Belinda Giorgetti, Pierpaolo Stripoli, Anna Rita Bonfigli, Massimiliano Fedecostante, Fabio Salvi, Adolfo Pansoni, Mauro Provinciali, Fabrizia Lattanzio

**Affiliations:** 1Advanced Technology Center for Aging Research, IRCCS INRCA, 60121 Ancona, Italy; 2Geriatria, Accettazione geriatrica e Centro di ricerca per l’invecchiamento, IRCCS INRCA, 60127 Ancona, Italy; 3Clinical Laboratory and Molecular Diagnostic, Italian National Research Center on Aging, IRCCS INRCA, 60127 Ancona, Italy; 4Center of Clinical Pathology and Innovative Therapy, IRCCS INRCA, 60121 Ancona, Italy; 5Center for Neurobiology of Aging, Scientific Technological Area, IRCCS INRCA, 60121 Ancona, Italy; 6Scientific Direction, IRCCS INRCA, 60124 Ancona, Italy; 7Emergency Department, IRCSS INRCA, 60027 Osimo, Italy

**Keywords:** sensitivity, specificity, rapid diagnostic test, VitaPCR, SARS-CoV-2

## Abstract

(1) Background: During the COVID-19 pandemic, rapid and reliable diagnostic tools are needed for detecting SARS-CoV-2 infection in urgent cases at admission to the hospital. We aimed to assess the performances of the rapid molecular VitaPCR™ test (Menarini Diagnostics) in a sample of older adults admitted to the Emergency Department of two Italian hospitals (2) Methods: The comparison between the rapid VitaPCR™ and the RT-PCR was performed in 1695 samples. Two naso-pharyngeal swab samplings from each individual were obtained and processed using the VitaPCR™ and the RT-PCR for the detection of SARS-CoV-2 (3) Results: VitaPCR™ exhibited good precision (<3% CV) and an almost perfect overall agreement (Cohen’s K = 0.90) with the RT-PCR. The limit of detection of the VitaPCR™ was 4.1 copies/µL. Compared to the RT-PCR, the sensitivity, the specificity, and the positive and negative predictive values of VitaPCR™ were 83.4%, 99.9%, 99.2% and 98.3%, respectively (4) Conclusions: The VitaPCR™ showed similar sensitivity and specificity to other molecular-based rapid tests. This study suggests that the VitaPCR™ can allow the rapid management of patients within the Emergency Department. Nevertheless, it is advisable to obtain a negative result by a RT-PCR assay before admitting a patient to a regular ward.

## 1. Introduction

Coronavirus disease-2019 (COVID-19), the illness caused by the severe acute respiratory syndrome coronavirus 2 (SARS-CoV-2) virus, had a worldwide spread and affected both inpatient and outpatient healthcare institutions [1,2]. Approximately 430 million cases of confirmed SARS-CoV-2 infection and 5.9 million deaths from COVID-19 have been reported to the World Health Organization (WHO) [3].

SARS-CoV-2 testing has been critical for identifying cases of COVID-19, reducing transmission, and taking public health infection-control measures, especially in the hospital setting [4]. Effective interventions to limit the nosocomial transmission of the SARS-CoV-2 are pivotal to maintain healthcare capacities during pandemic waves. However, despite the adoption of rigid infection control protocols, many hospitals experienced nosocomial transmission of the disease [5,6,7]. Therefore, a major concern for health systems is mitigating the spread of the virus within the hospital setting, both to uninfected patients and to healthcare workers. Currently, the laboratory tests, based on reverse transcriptase polymerase chain reaction (RT-PCR), are considered the gold standard to detect the SARS-CoV-2 infection. However, these methods require high costs for reagents and equipment, skilled personnel, and several hours to produce a result, often not being available during the night. Many hospitals have implemented strategies to test patients directly in the Emergency Department (ED); this is by using Rapid Diagnostic Tests (RDTs) prior to admission to hospital units, with the goal of appropriately admitting COVID-19 positive patients to COVID-specific wards [8]. RDTs can provide a result ‘while you are waiting’, ideally within one hour of providing a sample. This could help to isolate people early and reduce the spread of infection.

RDTs for acute SARS-CoV-2 infection can be performed with either molecular nucleic acid amplification tests (NAATs) or antigen-based assays [9,10]. Considering that antigen tests are able to detect the infection only after the beginning of symptoms (3–4 days after the virus replication), NAATs should be preferred for detecting SARS-CoV-2 positivity in individuals presenting at ED, prior to hospital admission [11,12].

Several molecular RDTs allowing a diagnosis in less than an hour have been evaluated. Of these, the two fastest tests (Abbott ID NOW and Mesa Accula), with less than 30 min between the sampling and result, showed good to poor diagnostic performance [11,13,14,15]. The Cepheid Xpert Xpress SARS-CoV-2 assay has been shown to be a valuable tool with a run-time of 45–50 min, with hands on time limited to 2–3 min [16]. An interesting RDT is represented by the VitaPCR™ (Menarini Diagnostics), which was found to be highly sensitive, enabling the delivery of results in less than half an hour [17,18]. VitaPCR™ is a molecular point-of-care platform, based on fast RT-PCR for rapid molecular detection, and directly from oro and nasopharingeal swabs of SARSCoV-2 and other respiratory viruses. The system delivers actionable results in only 20 min. All assays running on the VitaPCR system are based on fast RT-PCR with multicolor fluorescence detection in up to four different channels. Sample preparation includes single-step lysis and extraction solutions to allow an efficient and rapid release of nucleic acids. The VitaPCR™ assay includes three detection systems: (1) one targeting the human β-globin gene, to check the quality of DNA extracts; (2) a second targeting a specific sequence on the nucleocapside N-encoding gene; (3) and a third targeting a conserved sequence common to SARS-CoV-2, SARS-CoV and the SARS-like bat coronavirus, also located on the N-encoding gene. An overview of the mentioned instruments, with advantages and drawbacks, has been provided in the Appendix A.

The objective of this study was to determine the diagnostic performance of the VitaPCR™ COVID-19 assay for the detection of SARS-CoV-2; this was with respect to the current gold standard of RT-PCR testing in elderly patients at the time of admission into the ED of two geriatric hospitals, both located in the Marche Region, Italy. In particular, VitaPCR™ sensitivity, specificity, positive and negative predictive values (PPV, NPV), as well as precision, concordance and overall agreement with the laboratory-based RT-PCR were determined.

## 2. Materials and Methods

### 2.1. Study Design

A retrospective analysis was conducted to evaluate the accuracy of the VitaPCR™ RDT (Menarini Diagnostics, Florence, Italy) for the detection of SARS-CoV-2, with respect to the laboratory-based RT-PCR. The analysis included older adults (≥65 years) presenting at the EDs of two hospitals at the National Institute of Research and Care of Aging, IRCCS INRCA, located in Ancona and Osimo, Marche Region, Italy. We collected data from our electronic medical record during a 1-year period from May 2020 to May 2021. The study was approved Ethics Committee of the IRCCS INRCA, Ancona, Italy (reference number N.0037569/2022; 23/11/2022). The comparison analysis between the VitaPCR™ and the laboratory-based RT-PCR test was performed only on those patients who underwent two naso-pharyngeal swab samplings; these samplings were analyzed by the rapid VitaPCR™ method and by the laboratory-based RT-PCR equipment (reference methods) for the detection of SARS-CoV-2. Both the VitaPCR™ and the laboratory-based RT-PCR were installed at the IRCCS INRCA.

### 2.2. VitaPCR Instrument

For virus lysis and inactivation, the swab was discharged in the kit-provided collection buffer by stirring it 20 times. We allowed the lysis buffer to act for 5 min. Thirty μL of lysate were transferred to the tube containing the lyophilized PCR reagents and were mixed well by pipetting. Bubbles were avoided throughout the process. The tube was then introduced into the VitaPCR™ instrument in order to perform the analysis by RT-PCR. The results were obtained after 20 min of analysis.

The VitaPCR™ provided several outputs after the sample analysis: ND (not detected), P (positive), P SARS-like (positive for coronavirus gene), Indeterminate (gene not fully detected), Invalid.

### 2.3. Laboratory-Based RT-PCR Instruments

RNA extraction was performed by the automated Qiacube (Qiagen, Hilden, Germany) using the related RNA extraction kit (RNAeasy, Qiagen), and following the manufacturer’s instructions.

Our laboratory-based RT-PCR instruments were the following:
-the Rotor-Gene Q Real Time PCR System (Qiagen) for targeting the envelope protein E-encoding gene, the N gene and the SARS-like gene through the use of the multi-target Real Time PCR kit (Sacace Biotechnologies, Como, Italy)-the Aria Mix Real Time PCR System (Agilent Diagnostics, Santa Clara, CA, USA) for targeting the envelope protein E-encoding gene, the S gene and the RDRP gene through the use of the kit AB Analytica RQ 129 (Gene RDRP, Gene E and IC) and RQ 130 (Gene RDRP, Gene S, and IC).

All the analyses for the detection of SARS-CoV-2 were performed following the manufacturer’s instructions.

### 2.4. Valid Results

The criteria for a valid result with either VitaPCR or the laboratory RT-PCR were as follows:
-IC gene (for kit AB Analytica RQ 129 and kit SACACE) aligned with CT values provided by the manufacturer.-Endogen Control gene (for kit AB Analytica RQ 130 and kit Menarini VitaPCR) in line with the mean results of the batch.-for a positive control sample: CT values referred to the RDRP gene, E gene and N gene aligned with CT values provided by the manufacturer.-for a negative control sample: CT values referred to the RDRP gene, E gene and N gene not detected. -for an unknown sample: CT values referred to the IC gene aligned with the CT values provided by the manufacturer. CT values referred to the endogen control gene similar to those of the IC gene provided by the manufacturer. Curve fitting of the gene RDRP, gene E and gene N in line with those of positive control samples if “positive”, and below the limit of detection of the method (37 CT for VitaPCR, 37 CT for Aria Mix and 35 CT for Rotor Gene Q gene N) if “negative”.

### 2.5. Limit of Detection

Limit of detection (LoD) defines the lowest detectable concentration at which ≥95% of the replicates were positive. The in vitro-transcribed full length SARS-CoV-2 RNA (N gene, RDRP gene and E gene) positive control (PC) samples were diluted from 1:1 to 1:104, and assessed on both a VitaPCR™, Rotor-Gene Q and Aria Mix Real Time PCR System. RNA concentration was expressed as copies/µL and the mean Cycles Threshold (CT) value.

### 2.6. Precision

Precision of the VitaPCR™, expressed as the Coefficient of Variation (CV%), was calculated as the ratio of the standard deviation to the mean. PC samples (*n* = 20) were assessed in order to obtain the CV%.

### 2.7. Concordance

The concordance between the rapid test (VitaPCR™) and the gold standard test (Rotor Gene Q) was evaluated by calculating Lin’s concordance coefficient. The CT obtained for each PC dilution (from 1:1 to 1:104) from both the VitaPCR™ and the Rotor-Gene Q were used and compared for the calculation of Lin’s concordance. MedCalc software was used to measure Lin’s concordance.

### 2.8. Sensitivity and Specificity

All the individuals analyzed by both the VitaPCR™ and the Aria Mix Real Time PCR System (Agilent Diagnostics) were included for the sensitivity and specificity determination. False negative and false positive results were also repeated by the Rotor-Gene Q Real Time PCR System (Qiagen).

Sensitivity was referred to as the probability of obtaining a positive result from the Vita PCR test when the disease was present (assessed by a positive result from RT-PCR).

Specificity was referred to as the probability of obtaining a negative result from Vita PCR when the disease was absent (assessed by a negative result from RT-PCR).

The positive predictive value was assessed as the probability that subjects, with a positive screening test, truly had the disease. The negative predictive value was calculated as the probability that subjects with a negative screening test truly did not have the disease.

### 2.9. Overall Agreement

The overall agreement between VitaPCR™ and the Aria Mix Real Time PCR System (Agilent Diagnostics) outputs was estimated by determining the Cohen’s K coefficient, which is a metric often used to assess the agreement between two raters. From both the Aria Mix Real Time PCR System and VitaPCR™, only P (Positive) and ND (Not Detected) outputs were considered. Idostatistics software was used to measure the Cohen’s K coefficient.

## 3. Results

### 3.1. Participants

During the study period, there were 3050 eligible patients admitted to a ED, of whom 1695 had valid results for both the VitaPCR™ and the laboratory-based RT-PCR (Figure 1). *n* = 1355 patients that had VitaPCR™ results but not RT-PCR results, because these patients could be discharged after a few hours from the ED; after a resulting “ND” by the VitaPCR™, an analysis by the RT-PCR test was not performed (Figure 1). These patients were not included in the analyses.

In total, 42 samples were found “invalid” by the VitaPCR™ were repeated by either VitaPCR™ or by both a VitaPCR™ and RT-PCR test, depending on the hospital admission need. In total, 22 of these were re-analyzed by both a VitaPCR™ and RT-PCR test, resulting in “ND”. These patients were included in the analysis within the group of patients resulting in “ND”. In total, 20 patients with only a negative result via VitaPCR™ were excluded from the analysis. 

In total, 131 patients tested positive via both the VitaPCR™ and the RT-PCR test, for a COVID-19 prevalence of 7.7%. Participant flow is outlined in Figure 1. Patients had a mean age of 82.6 ± 12.6 years, and 58% were women.

### 3.2. Limit of Detection

By assessing serial dilutions of the Menarini PC sample on both the VitaPCR™ and the Rotor-Gene Q, the LoD comparison of both instruments was performed through the calculation of the Cycle Threshold (CT) of each sample (Figure 2a). In an RT-PCR assay, a positive reaction is detected by the accumulation of a fluorescent signal. The CT is defined as the number of cycles required for the fluorescent signal to cross the threshold (i.e., exceeds the background level). CT levels are inversely proportional to the amount of target nucleic acid in the sample (i.e., the lower the CT level, the greater the amount of target nucleic acid in the sample). The LoD of the VitaPCR™ was approximately the same as that estimated by the manufacturer (LoD = 4.1 copies/µL at 37 CT).

A mean difference of 3 CT, between the VitaPCR™ (higher) and the Rotor Gene Q at each dilution, was found (Figure 2a). This fact implied that values > 34 CT, via laboratory based RT-PCR, could not be recognized as positive by the VitaPCR™; this is because they would have a CT value > 37 (over the LoD).

By assessing serial dilutions of the Sacace PC sample, the LoD (gene N) of the Rotor-Gene Q was estimated = 1.8 copies/µL (Figure 2b).

By assessing serial dilutions of the RQ-129 PCR PC sample, the LoD (gene RDRP) of the Aria Mix Real Time PCR System was estimated = 2.5 copies/µL (Figure 2c).

### 3.3. Precision

Precision of the VitaPCR™ was very good at under the 3% of the CV, also in diluted samples (Table 1).

### 3.4. Concordance

The concordance between VitaPCR™ and the Rotor-Gene Q (Cycle Threshold comparison) was poor (Lin’s concordance coefficient = 0.70; CI 95% 0.44–0.85).

### 3.5. Sensitivity and Specificity

Among the 1695 samples included in the study, 131 were positive and 1495 were negative, according to both the VitaPCR™ and the laboratory-based RT-PCR (Table 2); meanwhile, there were 36 false negatives and 1 false positive. All the results obtained by the VitaPCR™ and the laboratory-based RT-PCR are displayed in Table 2 and in Figure 1.

False negative and false positive samples were analyzed by both The Aria Mix Real Time PCR System and the Rotor gene Q Real Time PCR System. In particular, concerning false negative samples, the following were found: (a)*n* = 6/135 (4.4%) samples showed a mean CT < 30 (18 copies/µL); *n* = 12/14 (85.7%) samples showed a mean CT 31–34 (4.1–18 copies/µL) and 18/18 (100%) showed a mean CT > 34 (<4.1 copies/µL).(b)*n* = 10 patients were at the beginning of the infection (first diagnosis of SARS-CoV-2), *n* = 15 patients were at the end of the infection (final hospitalization) and *n* = 11 patients were analyzed only one time at IRCCS INRCA.(c)The mean CT obtained by the Aria Mix Real Time PCR System was equal to (mean ± SD): (i) 32 ± 4 and 33 ± 4 for gene RDRP and gene E, respectively (obtained by the Real Quality kit 129), and (ii) 32 ± 5 and 34 ± 5 for gene RDRP and gene S, respectively (obtained by the Real Quality kit 130).(d)The mean CT obtained by the Rotor gene Q Real Time PCR System was equal to 33 ± 5 and 34 ± 4 for the gene N and the gene E, respectively, by using the multi-target Real Time PCR kit (SACACE).

Both instruments confirmed the analysis “ND” for the unique false positive sample.

Considering that the LoD of the Aria Mix Real Time PCR System was = 2.5 copies/µL at 36 CT (Figure 3), *n* = 10 false negative samples with a CT > 36 were excluded from the calculation of sensitivity, specificity and overall agreement.

Sensitivity decreased concomitantly with the decrease in the nasopharyngeal viral load (Figure 3). The sensitivity, the specificity, the positive, and the negative predictive values of the VitaPCR™ were 83.4% (CI 95% 81.5–85.2), 99.9% (CI 95% 99.6–100), 99.2% (CI 95% 98.7–99.6) and 98.3% (CI 95% 97.5–98.8), respectively.

### 3.6. Overall Agreement

An almost perfect agreement between VitaPCR™ and the laboratory-based RT-PCR outputs was detected, as indicated by a Cohen’s Kappa coefficient of 0.90 (CI95% 0.86–0.94).

## 4. Discussion

SARS-CoV-2 is a highly contagious and rapidly evolving infectious agent. For this reason, it is important to reliably identify infected patients, in order to reduce the diffusion of the virus, particularly in the hospital setting. To date, laboratory-based RT-PCR methods are considered the gold standard for COVID-19 detection, although they have several practical limitations. Consequently, alternative diagnostic methods have been developed, both for alleviating the pressure on laboratories and healthcare facilities, and for expanding the testing capacity to enable large-scale screening and ensure a timely therapeutic intervention. RDTs, demonstrated in multiple studies to have an optimal specificity but lower sensitivity, suggest that the use of available tests should be properly evaluated in the clinical setting [19].

This study analyzed the precision and accuracy of the VitaPCR™ (Menarini Diagnostics) molecular RDT in older adults (mean age 82.6 y) who presented at the EDs of two hospitals of the IRCCS INRCA, located at Ancona and Osimo (Italy), from May 2020 to May 2021. The results showed an almost perfect agreement (Cohen’s k = 0.90) between VitaPCR™ and the laboratory-based RT-PCR, an excellent specificity (almost 100%) but a less than optimal sensitivity (83.4%). This latter figure is slightly lower than that found by Fournier et al. (2020) and by Fitoussi et al. (2021) in other studies with the same RDT but with a much lower sample size [17,18]. The lower VitaPCR™ sensitivity observed in this study might be explained by the lower LoD of our RT-PCR (2.1 copies/µL). The LoD of the RT-PCR (gold standard) is much lower, and probability of obtaining false negative results is much higher via VitaPCR™. However, in line with the findings reported by Fitoussi et al., we observed that VitaPCR™ sensitivity decreased concomitantly with the decrease in the nasopharyngeal viral load [17] (Figure 3). Indeed, it is not surprising to observe that patients with a viral load lower than 4.1 copies/µL (the VitaPCR™ LoD) were incorrectly identified as negative, representing false negative cases (see Figure 2). Nevertheless, half of the very few false negative samples had viral loads above the limit of detection (LoD). There might be several reasons for this. Previous research has identified a variety of potential causes for false negative results, including improper materials or sampling techniques, poor preservation or prolonged transportation times [20,21]. Therefore, it is possible that any of these factors may have affected the accuracy of the VitaPCR™ analysis.

Interestingly, the VitaPCR™ showed a higher mean value of three cycles than that one of Rotor Gene Q at each dilution (Figure 2a). No evidence supporting this difference was found in the literature. This difference could be explained, as suggested by Wee et al., by a reduction in the amplification efficiency; this is a common concern in DIRECT-PCR (without RNA extraction) from crude samples (e.g., respiratory samples, blood and serum) [22]. The presence of PCR inhibitors, such as mucin and proteins, can decrease the sensitivity and accuracy of pathogen detection through interfering with polymerase activity, degradation of nucleic acids and efficient cell lysis [23]. However, the results obtained by Fournier et al. [18], which obtained lower mean CT values by VitaPCR™ with respect to those of the RT-PCR for each sample analyzed, are in contrast with this finding. This discrepancy, in regard to the results of our study, could be due to the differing variables, as follows: (i) laboratory-based RT-PCR instrument used by Fournier et al. (KingFisher™ Flex system Thermo Fisher Scientific), (ii) settings of the threshold and (iii) amplification protocols.

Considering the potential differences between/within RDTs and laboratory-based RT-PCR instrumental settings and amplification protocols, this study warns against comparing CT results; however, it suggests comparing the outcomes by the measure of the copies/µL.

Concerning the false negative individuals, it was observed that one-third of them were at the beginning of the infection, whereas another third were at the end. The last third were patients with only one spot analysis. These findings would alert all the healthcare workers to the risk of the virus spreading within the hospital; this is if a negative result with the VitaPCR™ could not be confirmed by a laboratory-based NAAT.

The overall sensitivity of the VitaPCR™ was similar to that of the ID NOW SARS-CoV-2 assay (Abbott), estimated by Tu et al. to be = 84% (95% CI 55–96%) and by Ramachandran et al. to be = 85.1% (95% CI 75.0% to 92.3%) [13,24]. Conversely, the VitaPCR™ demonstrated a lower sensitivity than that of the Xpert Xpress assay (Cepheid), which showed a pooled sensitivity = 0.99 (95% CI, 0.97 to 0.99) [25], and that of the CovidNudge test for SARS-CoV-2, which displayed a 94% sensitivity when compared with the standard laboratory-based RT-PCR [24]. However, only the Xpert Xpress assay demonstrated excellent results in several studies. Finally, the VitaPCR™ displayed a higher sensitivity than that observed by the Accula SARS-CoV-2 test (Mesa Biotech) when compared to the laboratory-based RT-PCR [15]. The Accula showed excellent negative agreement, whereas the positive agreement was low for samples with a low viral load [15]. Regarding the time needed to deliver the results, the VitaPCR™ was one of the fastest, taking only 20 min to process each sample, preceded only by the ID NOW (13 min). Conversely, instead of using CovidNudge, whose array consists of seven viral targets [26], the possibility of measuring only one gene for the detection of SARS-CoV-2 (gene N), via VitaPCR™, could affect the identification of SARS-CoV-2 if a mutation of the gene N is present. An overview of the mentioned instruments with advantages and drawbacks has been provided in the Appendix A.

### Strengths and Limitations

This study does not provide any comparison between the VitaPCR™ RDT and the “near patient antigen testing”; this is because, in the inquired period (from May 2020 to May 2021), only the laboratory-based NAATs were considered reliable diagnostic tests for the patient’s management in the ED. While our study includes only older adults (mean age of 82.6 years), which may limit its generalizability, on the other hand, our study represents one of the confirmations of how this RDT might be successfully applied to the rapid management of older individuals in an ED. During the pandemic, not all of the samples, analyzed via VitaPCR™, were replicated by the laboratory-based RT-PCR; this is because patients with a negative VitaPCR™ result, and who did not need to be admitted, were not subjected to the RT-PCR confirmation. Indeed, in this study, we only included individuals with two nasopharyngeal swabs assessed by both VitaPCR™ and laboratory-based RT-PCR. Clinical data were not available for all the individuals at the ED, thus affecting potential evaluations of the VitaPCR™ sensitivity, in relation to clinical symptoms. However, considering the results obtained in previous studies [27,28], which found similar CT values in asymptomatic patients compared with those in symptomatic patients, the accuracy of VitaPCR™, evaluated in this study, should not suffer due to this lack of clinical data.

## 5. Conclusions

Considering our results, we suggest that VitaPCR™ might be useful as a first-line screening test in the ED for older adults, to quickly identify and isolate positive patients. On the other hand, if the patient needs to be admitted to a hospital ward, a negative result should be confirmed by a laboratory-based NAAT.

## Figures and Tables

**Figure 1 viruses-15-00189-f001:**
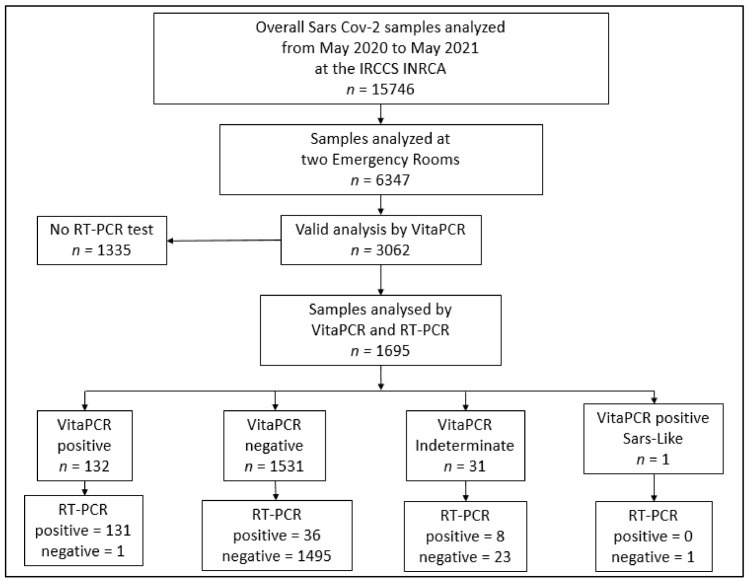
Flow chart of patients’ selection. Flowchart of the selection of samples analyzed for SARS-CoV-2 with both VitaPCR™ and RT-PCR methods. Abbreviations: RT-PCR, reverse transcription polymerase chain reaction; ED, Emergency Department.

**Figure 2 viruses-15-00189-f002:**
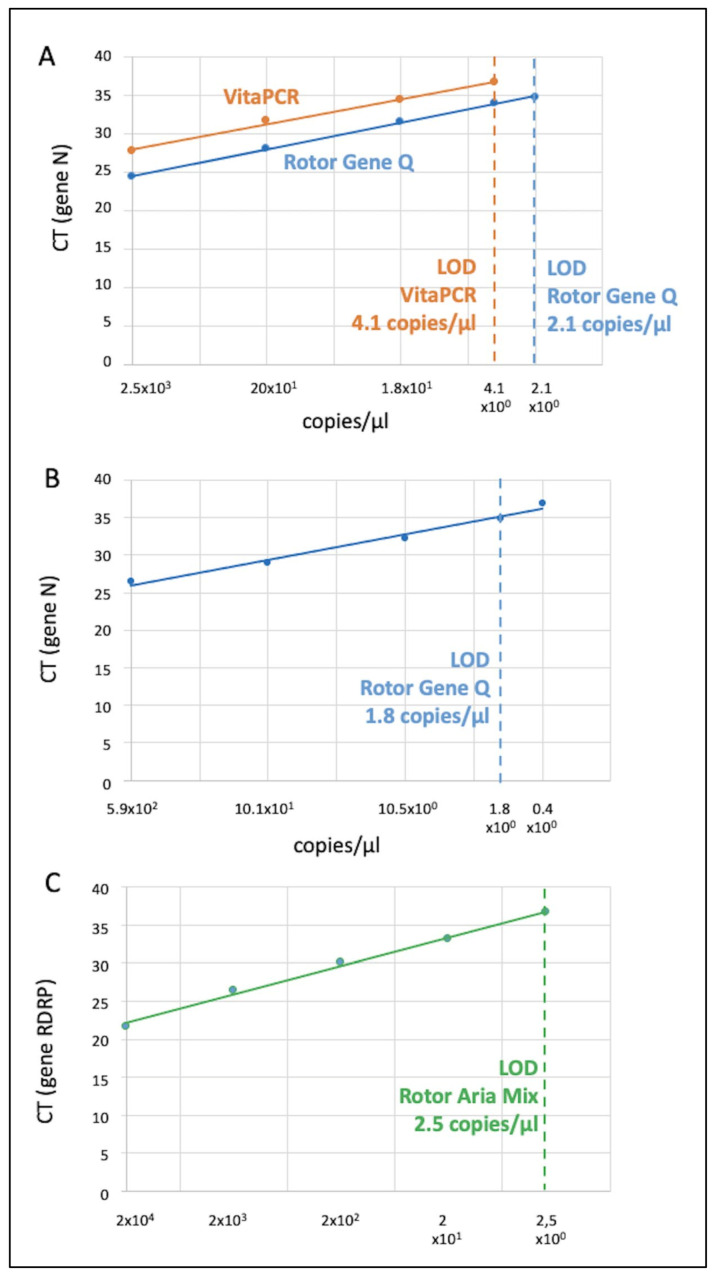
Limit of detection of the VitaPCR™, the Rotor Gene Q and the Aria Mix. The figure represents the LoD of (**A**) the VitaPCR™ and the Rotor Gene Q (Gene N) by the analysis of the Menarini Positive Control sample (and the same diluted samples); (**B**) the Rotor Gene Q (Gene N) via the analysis of the Sacace RT-PCR Positive Control sample; (**C**) the Aria Mix RT-PCR System (Gene RDRP) via the analysis of the Real Quality 129 Positive Control sample. Data of RNA copies/µL were log-converted (only for the figure) to obtain a linear fitting.

**Figure 3 viruses-15-00189-f003:**
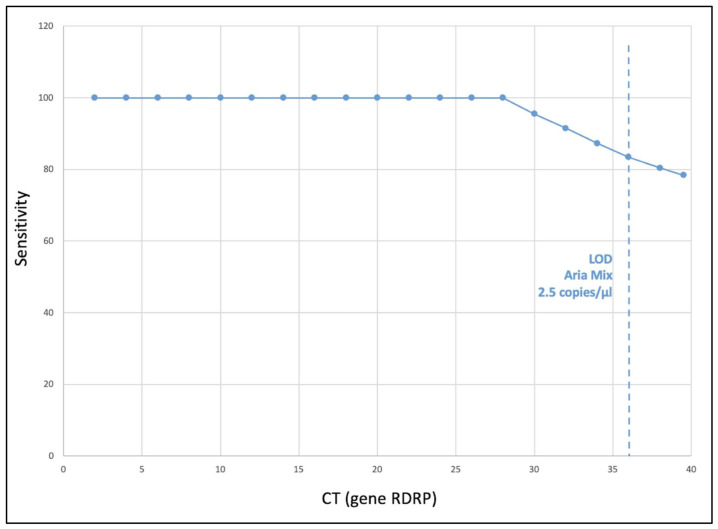
Sensitivity of the VitaPCR™. The figure represents the sensitivity of VitaPCR™ (false negative rate in relation to the overall positive) in relation to the CT obtained by the laboratory-based RT-PCR Aria Mix. Considering that the LoD of the Aria Mix was 2.5 copies/µL (36CT), the sensitivity of VitaPCR™ was calculated, including results up to 36 CT.

**Table 1 viruses-15-00189-t001:** Precision of the VitaPCR™.

VitaPCR™ PC Dilutions	Mean CT	SD CT	CV	CV%
**1:1**	27.7	0.58	0.021	2.1
**1:10**	31.7	0.55	0.017	1.7
**1:100**	34.3	0.85	0.025	2.5
**1:500**	36.7	1.06	0.029	2.9

PC = positive control; CT = cycle threshold; SD = standard deviation; CV = coefficient of variance.

**Table 2 viruses-15-00189-t002:** VitaPCR™ results in comparison with the gold standard (RT-PCR).

		RT-PCR			
		ND		POSITIVE	
		N	%	N	%
**VitaPCR™**	**ND (*n* = 1531)**	1495	97.6%	36	2.4%
	**P (*n* = 132)**	1	0.8%	131	99.2%
	**I (*n* = 31)**	23	74.2%	8	25.8%
	**SARS (*n* = 1)**	1	100%	0	0%

ND = not detected; P = positive; D = Indeterminate sample; Sars = positive for Sars-Like Virus.

## Data Availability

The data presented in this study are available on request from the corresponding author.

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
