# Peer review of "Sensibility and Specificity of the VitaPCR™ SARS-CoV-2 Assay for the Rapid Diagnosis of COVID-19 in Older Adults in the Emergency Department"

_viruses, 2023, doi:10.3390/v15010189_

Round 1

Reviewer 1 Report

In this study, authors mainly compared the diagnostic performance of the VitaPCR assay with the traditional RT-PCR testing, including its precision, sensitivity, specificity, positive and negative predictive values. This study indeed brings up an important issue during the post-SARS-CoV-2 era, which is to quickly and economically examine the SARS-CoV-2 positivity for individuals at emergency department prior the hospital admission. The authors finally concluded that VitaPCR showed sensitivity, specificity and overall agreement with the traditional RT-PCR. Generally, the topic is of interest for a broad audience. However, several issues need to be addressed prior publication.

1. Line 75-77, authors only cited two references, 17 and 18, for the introduction of VitaPCR. Obviously, this is not enough. More information and details, such as the experimental principle, should be provided in the introduction part to facilitate the readers’ understanding.

2. Line 188-190, the authors stated that, values >34 CT by laboratory-based RT-PCR could not be recognized as positive by the VitaPCR because they would have a CT value >37 (over the LoD). If it is true, many positive samples will be considered as negative by VitaPCR.

Would this be a problem? Actually, in shown in Figure 1, 36/1531 (2.35%) samples were positive by conventional RT-PCR assay, but were negative by VitaPCR. Therefore, this might potentially be a problem.

3. In abstract, the authors mentioned that “the limit of detection of the Vita PCR was 4.1 copies/ul.” How did the authors estimate this number? I did not find any explanation in the text. Moreover, for those 36 RT-PCR-positive, but VitaPCR-negative samples, what are their viral copies? Are the viral copy numbers from those samples all below 4.1 copies/ul? Maybe more original data for Figure 1 needs to be presented.

4. The text in Figure 1 is too small to follow.

5. Line 187, the definition and meaning of CT value needs to be provided.

6. In the figure legend of Figure 2, the “a)”, “b)”, and “c)” should be modified as “A)”, “B)”, and “C)”.

7. Please change all “SARS-Cov-2” in the manuscript to “SARS-CoV-2”.

8. The authors compared several RDTs in the Introduction part. Maybe a table could be added here to elucidate the advantages and drawbacks for each different RDT method.

Author Response

Reviewer 1

In this study, authors mainly compared the diagnostic performance of the VitaPCR assay with the traditional RT-PCR testing, including its precision, sensitivity, specificity, positive and negative predictive values. This study indeed brings up an important issue during the post-SARS-CoV-2 era, which is to quickly and economically examine the SARS-CoV-2 positivity for individuals at emergency department prior the hospital admission. The authors finally concluded that VitaPCR showed sensitivity, specificity and overall agreement with the traditional RT-PCR. Generally, the topic is of interest for a broad audience. However, several issues need to be addressed prior publication.

  1. Line 75-77, authors only cited two references, 17 and 18, for the introduction of VitaPCR. Obviously, this is not enough. More information and details, such as the experimental principle, should be provided in the introduction part to facilitate the readers’ understanding.

Following the reviewer suggestion, both the principle and the operational characteristics of the VitaPCR were described in the Introduction section (Line 77-88)

  1. Line 188-190, the authors stated that, values >34 CT by laboratory-based RT-PCR could not be recognized as positive by the VitaPCR because they would have a CT value >37 (over the LoD). If it is true, many positive samples will be considered as negative by VitaPCR.

Would this be a problem? Actually, in shown in Figure 1, 36/1531 (2.35%) samples were positive by conventional RT-PCR assay, but were negative by VitaPCR. Therefore, this might potentially be a problem.

The issue raised by the reviewer is indeed the main point of the manuscript. Our study demonstrated that the VitaPCR is an effective tool for identifying SARS-CoV-2 positive patients with a viral infection of more than 4 copies/microliter per sample. However, the VitaPCR might fail to correctly identify patients who are in the early stages of infection with very low levels of the virus. If these patients are admitted to a hospital without confirmation from a laboratory RT-PCR test, they may pose a risk of infecting both healthcare workers and other patients. This is because SARS-CoV-2 can rapidly multiply, and a patient who tests negative with the VitaPCR but has low levels of the virus may become positive a few hours later. Therefore, it is recommended to re-test patients who test negative with the VitaPCR using the laboratory RT-PCR before admitting them to the hospital. This recommendation is provided both in the Abstract and in the conclusions of the manuscript.

  1. In abstract, the authors mentioned that “the limit of detection of the Vita PCR was 4.1 copies/ul.” How did the authors estimate this number? I did not find any explanation in the text.

As described in the methods section we used different analytical procedures on basis of the instrument used:

- with the Aria Mix Real Time PCR System (Agilent Diagnostics) two analytical kits from AB Analytica were used: the RQ 129 for targeting Gene RDRP, Gene E and IC and the RQ 130 for Gene RDRP, Gene S, and IC.

- with the Rotor-Gene Q Real Time PCR System (Qiagen) the multi-target Real Time PCR kit (SACACE) was used for targeting the envelope protein E-encoding gene, the N gene and the SARS-like gene.

All the analyses for the SARS-CoV-2 detection were performed following the manufacturer’s instructions.

The kit from AB Analytica was the only one that declared the number of the viral copies related to the positive standard sample. From this sample, we generated a calibration curve for the calculation of the viral copies of the kit SACACE. After obtaining these results, we calculated also the viral copies for the standard positive sample of the VitaPCR. By this way we were able to obtain the viral copies for all the positive patients.

Moreover, for those 36 RT-PCR-positive, but VitaPCR-negative samples, what are their viral copies? Are the viral copy numbers from those samples all below 4.1 copies/ul? Maybe more original data for Figure 1 needs to be presented.

We would like to thank the reviewer for this question. We improved the results section including the description of the false negative results in the line 235: “n=6/135 (4.4%) samples showed a mean CT<30 (18 copies/µl); n=13/14 (92.9%) samples showed a mean CT 31-34 (4.1-18 copies/µl) and 18/18 (100%) showed a mean CT>34 (<4.1 copies/µl)”. 

Moreover, considering that half of the false positive samples were above the LoD, the following explication was added to the Discussion section: However, in line with those reported by Fitoussi et al, we observed that VitaPCR™ sensitivity decreased concomitantly with the decrease of nasopharyngeal viral load [17] (Figure 3). Indeed, it is not surprising to observe that patients with a viral load lower than 4.1 copies/µl (the VitaPCR™ LoD) were incorrectly identified as negative ones, representing false negative cases (see Figure 2). Nevertheless, half of the very few false negative samples had viral loads above the limit of detection (LoD). There might be several reasons for this. Previous research has identified a variety of potential causes for false negative results, including improper materials or sampling techniques, poor preservation or prolonged transportation times [20-21]. Therefore, it is possible that any of these factors may have affected the accuracy of the VitaPCR™ analysis. 

  1. The text in Figure 1 is too small to follow.

The text was modified as suggested.

  1. Line 187, the definition and meaning of CT value needs to be provided.

Following the reviewer's suggestion the following paragraph was added:

By assessing serial dilutions of the Menarini PC sample on both the VitaPCR™ and the Rotor-Gene Q, the LoD comparison of both Instruments was performed through the calculation of Cycle Threshold (CT) of each sample (Figure 2a). In an RT PCR assay a positive reaction is detected by accumulation of a fluorescent signal. The CT is defined as the number of cycles required for the fluorescent signal to cross the threshold (i.e. to exceed the background level). Ct levels are inversely proportional to the amount of target nucleic acid in the sample (ie the lower the Ct level the greater the amount of target nucleic acid in the sample).

  1. In the figure legend of Figure 2, the “a)”, “b)”, and “c)” should be modified as “A)”, “B)”, and “C)”.

The figure legend was modified as suggested.

  1. Please change all “SARS-Cov-2” in the manuscript to “SARS-CoV-2”.

The term was modified as suggested.

  1. The authors compared several RDTs in the Introduction part. Maybe a table could be added here to elucidate the advantages and drawbacks for each different RDT method.

Considering that we did not provided data with all the mentioned instruments, we included a comparative table in the supplementary file.

Reviewer 2 Report

Both fast and accurate detection of SARS-CoV-2 are indispensable for containment of this devastating pandemic. Many rapid molecular tests have emerged that, since speed is not always accompanied by accuracy of result with the current technoloy, need to be evaluated in terms of their reliability as accurate diagnostic means. The authors provide useful information about the usefulness of VitaPCR rapid test for correct diagnosis of SARS-CoV-2 and their results and conclusions are really sincere and important. I would suggest acceptance with minor revision of the manuscript. Points that may be revised, in my opinion, are listed below:

·         Line 69: Prior to hospital admission.

·         Line 71: …have been evaluated. (omit “in the literature”).

·         Line 73: …showed from poor to good diagnostic performance (not performances).

·         Line 111: …P SARS-like (positive for SARS-like) – I would not generally suggest the term “SARS-like”. In my opinion, the term common coronavirus gene may be better.

·         Which are the criteria for a valid result with either VitaPCR, or the other RT-PCR methods used for SARS-CoV-2 detection? For instance, detection of the housekeeping gene validated by a proper sigmoid fluorescence curve with fluorescence elevation of a certain number of units above the baseline and a Ct between a specific range of cycles, etc.

·         Also, which are the criteria for a positive, or negative result with either VitaPCR, or the other RT-PCR methods used? For instance, Ct values larger than 37 are considered to be negative by VitaPCR, etc.

·         Please, specify what does the VitaPCR interpretation software concludes as “indeterminate”, or, as the authors state, gene not fully detected.

·         The authors state that their study suggests to compare the outcomes by the measure of the copies/μl, instead of Ct values. Concerning the intra- and inter-assay variability of the performance of different RT-PCR assays, that sounds a reasonable means to use a consensus criterion for providing a specific and reliable SARS-CoV-2 detection result. But, considering that many RT-PCR protocols are performed on RNA extracted from different types and, most importantly, volumes of liquid transport media, such a criterion may be used with great caution, in my opinion. For example, the same swab may have a different copies/μl result when it is immersed in a 1ml transport medium tube than when it is immersed in a 3ml transport medium tube, not to mention different efficiencies between GuSCN-based, virus inactivating transport media and ordinary, non-virus-inactivating, Hank’s-type of transport media.

·         Which sample types has the VitaPCR been certified for? Since saliva is also a highly acceptable sample type for SARS-CoV-2 detection that is also far more easily collected, have the authors tried to test saliva directly, irrespective of whether VitaPCR has been certified for this type of sample? This is a question that, although important for diagnostics of SARS-CoV-2 in general, is perhaps not necessary to be answered regarding the specific objectives of the study and its final acceptance for publication.

Author Response

Reviewer 2

Both fast and accurate detection of SARS-CoV-2 are indispensable for containment of this devastating pandemic. Many rapid molecular tests have emerged that, since speed is not always accompanied by accuracy of result with the current technology, need to be evaluated in terms of their reliability as accurate diagnostic means. The authors provide useful information about the usefulness of VitaPCR rapid test for correct diagnosis of SARS-CoV-2 and their results and conclusions are really sincere and important. I would suggest acceptance with minor revision of the manuscript. Points that may be revised, in my opinion, are listed below:

  • Line 69: Prior to hospital admission.
  • Line 71: …have been evaluated. (omit “in the literature”).
  • Line 73: …showed from poor to good diagnostic performance (not performances).
  • Line 111: …P SARS-like (positive for SARS-like) – I would not generally suggest the term “SARS-like”. In my opinion, the term common coronavirus gene may be better.

The text was modified following the reviewer's suggestions.

  • Which are the criteria for a valid result with either VitaPCR, or the other RT-PCR methods used for SARS-CoV-2 detection? For instance, detection of the housekeeping gene validated by a proper sigmoid fluorescence curve with fluorescence elevation of a certain number of units above the baseline and a Ct between a specific range of cycles, etc.

Following the reviewer's suggestion, we provided a dedicated chapter for the description of the criteria used for a valid result.

2.4. Valid results

The criteria for a valid result with either VitaPCR or the laboratory RT-PCR were:

- IC gene (for kit AB Analytica RQ 129 and kit SACACE) aligned with CT values provided by the manufacturer.

- Endogen Control gene (for kit AB Analytica RQ 130 and kit Menarini VitaPCR) in line with the mean results of the batch.

- for a positive control sample: CT values referred to the RDRP gene, E gene and N gene aligned with CT values provided by the manufacturer.

- for a negative control sample: CT values referred to the RDRP gene, gene E gene and N gene not detected.

- for an unknown sample: CT values referred to the IC gene aligned with the CT values provided by the manufacturer. CT values referred to the endogen control gene similar to those of the IC gene provided by the manufacturer. Curve fitting of the gene RDRP, gene E and gene N in line with those of positive control samples if “positive” and below the Limit of detection of the method (37 CT for VitaPCR, 37 CT for Aria Mix and 35 CT for Rotor Gene Q gene N) if “negative”.

  • Also, which are the criteria for a positive, or negative result with either VitaPCR, or the other RT-PCR methods used? For instance, Ct values larger than 37 are considered to be negative by VitaPCR, etc.

Positive samples were those with CT values lower than the LoD (37 CT for VitaPCR, 37 CT for Aria Mix and 35 CT for Rotor Gene Q gene N) whereas negative ones were those with a CT value higher than the LoD.

  • Please, specify what does the VitaPCR interpretation software concludes as “indeterminate”, or, as the authors state, gene not fully detected.

When the VitaPCR gives an output declared “Indeterminate”, this could mean that: a) the sample amount (the obtained RNA) was lower than that required or b) the sample quality was lower than that required. In this case, the analysis was repeated in a new aliquot of the sample.

  • The authors state that their study suggests to compare the outcomes by the measure of the copies/μl, instead of Ct values. Concerning the intra- and inter-assay variability of the performance of different RT-PCR assays, that sounds a reasonable means to use a consensus criterion for providing a specific and reliable SARS-CoV-2 detection result. But, considering that many RT-PCR protocols are performed on RNA extracted from different types and, most importantly, volumes of liquid transport media, such a criterion may be used with great caution, in my opinion. For example, the same swab may have a different copies/μl result when it is immersed in a 1ml transport medium tube than when it is immersed in a 3ml transport medium tube, not to mention different efficiencies between GuSCN-based, virus inactivating transport media and ordinary, non-virus-inactivating, Hank’s-type of transport media.

This is true. In this respect, the Marche Region identified the Department of Virology at the regional hospital “Ospedali Riuniti” as the leading group for SARS-CoV-2 detection within the area of Marche, Italy. Our hospital was one of the few qualified from the leader hospital to perform SARS-CoV-2 analysis. For this reason, our procedures were certified.

  • Which sample types has the VitaPCR been certified for? Since saliva is also a highly acceptable sample type for SARS-CoV-2 detection that is also far more easily collected, have the authors tried to test saliva directly, irrespective of whether VitaPCR has been certified for this type of sample? This is a question that, although important for diagnostics of SARS-CoV-2 in general, is perhaps not necessary to be answered regarding the specific objectives of the study and its final acceptance for publication.

VitaPCR and the related analytical kit, are now certified for both oropharyngeal and nasopharyngeal swab samples. In 2020, the use of saliva samples for SARS-CoV-2 detection was not allowed by the Italian Ministry of Health (MoH). However, considering that now both the MoH and the manufacturer have validated the VitaPCR for also oropharyngeal swabs, another study could highlight the VitaPCR performance with this type of sample.